# Management of Tuberculosis Infection: Current Situation, Recent Developments and Operational Challenges

**DOI:** 10.3390/pathogens12030362

**Published:** 2023-02-21

**Authors:** Gino Agbota, Maryline Bonnet, Christian Lienhardt

**Affiliations:** 1University of Montpellier, TransVIHMI, Institut de Recherche pour le Développement (IRD), Institut National de la Santé et de la Recherche Médicale (INSERM), 34090 Montpellier, France; 2Institut de Recherches Cliniques du Bénin (IRCB), Abomey-Calavi, Benin; 3Epidemiology and Population Health, Department of Infectious Disease Epidemiology, London School of Hygiene and Tropical Medicine, London WC1E 7HT, UK

**Keywords:** TB infection, TB preventive therapy, TB high- and low-burden countries, cost-effectiveness, WHO recommendations

## Abstract

Tuberculosis infection (TBI) is defined as a state of infection in which individuals host live *Mycobacterium tuberculosis* with or without clinical signs of active TB. It is now understood as a dynamic process covering a spectrum of responses to infection resulting from the interaction between the TB bacilli and the host immune system. The global burden of TBI is about one-quarter of the world’s population, representing a reservoir of approximately 2 billion people. On average, 5–10% of people who are infected will develop TB disease over the course of their lives, but this risk is enhanced in a series of conditions, such as co-infection with HIV. The End-TB strategy promotes the programmatic management of TBI as a crucial endeavor to achieving global targets to end the TB epidemic. The current development of new diagnostic tests capable of discriminating between simple TBI and active TB, combined with novel short-course preventive treatments, will help achieve this goal. In this paper, we present the current situation and recent developments of management of TBI and the operational challenges.

## 1. Introduction

Tuberculosis (TB) is a major, yet preventable, global health problem [1]. According to the World Health Organization (WHO), in 2021, TB was among the top ten causes of death worldwide and the second leading cause of death from a single infectious agent, after COVID-19 [1]. The WHO estimated that 10.6 million people had developed active TB worldwide in 2021 and 1.6 million had died from it [1].

TB infection (TBI) is defined as a state of persistent immune response to stimulation by M. tuberculosis (*M.tb*) antigens with or without evidence of the clinical manifestation of active TB [2]. Most infected people have no clinical signs or symptoms and are not contagious, but they are at risk of progression to avert active TB, at which time they can contribute to transmission. On average, 5–10% of people who are infected will develop active TB over the course of their lives, with the highest risk in the first year after infection [3]. According to a recent re-estimation [4], the global burden of TBI is about one-quarter of the world’s population, representing a reservoir of approximately 2 billion people. People at extreme ages of life, people with compromised immune systems such as people living with HIV (PLHIV) or those who have a co-morbid condition such as malnutrition or diabetes, have a higher risk of progressing to active TB, with rates reaching as high as 10% per year among PLHIV [5].

In 2014, the World Health Assembly endorsed the ‘End-TB strategy’ [6] that aims to end the global TB epidemic by 2035, by reducing TB deaths by 95%, cutting down new cases by 90% and ensuring that no family faces catastrophic costs due to TB [6]. For the first time ever, this strategy promotes the management of TBI as a crucial endeavor to achieving global targets to end the TB epidemic. This led the WHO to update recommendations on TB prevention and control in 2019 so as to provide comprehensive guidance on the management of TBI in high- and low-TB-burden countries [7].

In this paper, we aim to provide an overview of the situation of TBI worldwide and present the current intervention strategies and recommendations for TB prevention in high- and low-TB-burden countries.

## 2. The Landscape of TBI Prevention in High- and Low-Burden Countries

### 2.1. The Evolving Concept of TBI

For decades, the outcome of infection by *M.tb* was represented as a bimodal distribution between latent and active TB (disease), based on the presence or absence of clinical symptoms and reaction to tuberculin skin testing. Since the beginning of the 21st century, TBI has been increasingly conceptualized as a dynamic and continuous spectrum of response to infection resulting from the interaction between TB bacilli and the host immune system [8]. Indeed, recent research suggests that asymptomatic people considered to have TBI might be distributed along a wide spectrum of infection states, where at one end, infection may have been eliminated, while at the other end, active TB may be already present but in a subclinical form, and between these two extremes, infection is variably controlled in a quiescent state [8,9]. Subsequently, as illustrated in Figure 1, the risk of developing active TB varies along this spectrum, depending on the balance of bacterial replication and efficacy of the host’s immune response [10]. Thus, at one end of the spectrum are the persons in whom infection is contained either through innate immune response or acquired T-cell immunity, while at the other end are persons with “incipient” TB [11], defined as infection with viable *M.tb* bacteria that has not yet caused clinical symptoms, radiographic abnormalities or microbiological evidence consistent with active TB, but who may be more likely to progress to active disease [12]. Incipient patients would be an attractive group to target TB preventive treatment (TPT) [13,14].

### 2.2. Global Burden

As mentioned above, about a quarter of the world’s population is estimated to be infected with *M.tb* [4,15]. Indeed, the global TBI prevalence was reported to be 23.0% in 2014 [4], and 24.8% (using interferon-gamma release assay) and 21.2% (using tuberculin skin test) in 2018 [15]. In the WHO regions, the estimated prevalence of TBI in 2018 was 36% in South-East Asia, 34% in Africa, 24% in the Eastern Mediterranean, 21% in the Western Pacific, 14% in the Americas and 12% in Europe [15]. The top three countries with the highest TBI burden from 1990 to 2019 were China, India and Indonesia [16]. 

Data on the prevalence of infection with strains that are resistant to first line anti-TB drugs (isoniazid and/or rifampicin) are not available, as infecting strains of *M.tb* cannot be isolated and tested for resistance. It has been estimated that about 11% of infections occurring in the world are due to an isoniazid-resistant *M.tb* strain [11]. A mathematical model, using cohorts tracked over time and historical data on the annual risk of infection, estimated that globally, 19.1 million people were latently infected with MDR-TB strains in 2014 [17]. MDR-TB strains accounted for 1.2% (95%CI 1.0–1.4) of the total burden of TBI overall, with a double burden in children younger than 15 years [17]. This has public health implications since either isoniazid monotherapy or combinations of isoniazid and rifamycin are not likely to be effective for TB prevention in these individuals.

### 2.3. Target Population

Not all individuals infected with *M.tb* ultimately progress to active TB, but the risk of the development of the disease varies according to individuals. It is therefore legitimate, from a public health point of view, to identify populations that are the most at risk of active TB for routine TB screening and provision of TPT. The proportion of individuals who will develop (over 2-years follow-up) active TB after TBI is particularly high in children and immunocompromised individuals [2]. Among children who have not received TPT, this proportion is 40–50% in infants younger than 12 months, decreases to 25% in children between 1 and 2 years of age, falls to 5–10% in school-aged children and then increases to 10–15% in adolescents [18]. On average, PLHIV TB co-infections have an 18-fold (95% CI: 15–21) higher risk of developing active TB than HIV-negative people [19]. This risk remains significantly high even after successful antiretroviral therapy (ART) [20,21]: PLHIV under ART were reported to have a 25% higher risk of active TB compared to HIV-negative people (active TB incidence rates (cases/100 person-year, [95% CI]) 2.70, [1.73–4.47] vs. 0.62, [0.58–0.65], respectively) [22]. For these reasons, the WHO recommends that PLHIV (>12 months of age) regardless of TB exposure, should be prioritized for receiving TPT after formally excluding active TB, even in the absence of TBI confirmation [23,24].

Household contacts (HHCs) of bacteriologically confirmed TB cases are another target population. A meta-analysis of prospective cohort studies of TB-exposed children showed that children with a positive TBI test not receiving TPT had a significantly higher cumulative incidence of active TB in the first 2 years of follow-up than children with a negative test result. The 2-year cumulative incidence decreased with the children’s age: 19.0% [95% CI 8.4–37.4] for <5 years; 9.0% [95% CI 4.0–20.0] for 5–9 years; 8.8% [95% CI 4.0–19.8] for 10–14 years; and 10.5% [95% CI 4.9–23.1] for 15–18 years [25]. The effectiveness of TPT was 63% (adjusted hazard ratio (HR) 0.37; 95% CI 0.30–0.47]) in all exposed children and 91% (adjusted HR 0.09; 95% CI 0.05–0.15) in those with a positive result for TBI testing [25]. Another systematic review of HHCs in high-TB-burden countries showed that all HHCs, regardless of age or TBI status, have a higher risk of progression to active TB than the general population (relative risk (RR): 24.7, 95% CI 14.2–43.0) [26]. For these reasons, the current WHO guidelines recommend the provision of TPT to all HHCs regardless of age after excluding active TB, with a priority to contacts < 5 years old and HIV-positive contacts > 5 years old, with no need to systematically confirm TBI [23,24].

Lastly, TPT should be considered in other groups at high risk of TBI and/or progression to active TB [2,23,24]. These are migrants from countries with a high TB burden; homeless persons; prisoners; illicit-drug users; patients receiving immunosuppressive therapy including TNFα inhibitors, long-term corticotherapy or undergoing hemodialysis; organ transplantation; elderly individuals; those with associated broncho-pulmonary disease, such as silicosis in miners; and health care workers.

### 2.4. TBI and COVID-19 Pandemic

The COVID-19 pandemic has severely affected the routine TB services due to the reduction in health systems’ capacity, the reassignment of health care workers to COVID-19 activities, the reallocation of TB diagnostic resources to COVID-19 and stigma associated with similarities in the symptoms related to TB and COVID-19 [19]. Implementation of lockdowns and strict quarantine measures resulted in delays in TB diagnosis, disruptions in contact-tracing activities and initiation of TPT and possibly increased transmission of TB within households [19]. The WHO reported an unprecedented drop in global TB case notifications between 2019 and 2020 (from 7.1 million to 5.8 million), with the largest reductions in the South-East Asia and Western Pacific regions [19]. In addition, the gap between the number of new TB duly notified cases and the estimated number of new cases has been shown to increase from 3.2 million in 2019 to 4.2 million people in 2021 [1,27]. In parallel, the number of people initiated on TPT decreased by 21% from 3.6 million in 2019 to 2.8 million in 2020 [19].

However, in the context of the disruption of essential TB services due to the COVID-19 pandemic, it is worth mentioning some successes in the implementation of innovative responses to TB [19,28]. These include multi-disease screening and testing strategies (GeneXpert for TB and COVID-19); automated chest X-ray interpretation and cough detection technologies at the point-of-care; and a shift from vertical to integrated approaches such as the use of community health workers to improve early TB case detection, diagnosis and care [19,27,28]; as well as the definition of a roadmap for access to the preventive screening of vulnerable populations in developed countries in line with the principles of universal health coverage [29]. Regarding TBI prevention, the COVID-19 pandemic has led to improvements in infection prevention and control within health systems, including the increased use of masks by patients and personal protective equipment by health care providers, resulting in a reduction in nosocomial transmission of both COVID-19 and TBI [30].

## 3. Recent Developments in TBI Management

### 3.1. TBI Diagnosis 

Identifying individuals with TBI at high risk of progression to active TB remains a challenge. There are two main diagnosis tests for TBI in use currently: the tuberculin skin test (TST) and the interferon-gamma release assays (IGRAs) [31]. Both tests are indirect tests based on the immune response to TB and do not directly assess the presence or viability of *M.tb* [32]. In addition, both tests cannot differentiate between different stages of TBI and have a low positive predictive value for progression to active disease [31], implying that a high number of people should be theoretically treated to prevent a case of active TB [33]. This may be problematic given the non-negligible risk of potential treatment-related adverse effects such as hepatotoxicity.

Among the newer TBI tests, the IGRA QuantiFERON-TB Plus (QFT-Plus) contains new antigens optimized for CD4+ and CD8+ T cell stimulation compared to the former QFT-Gold In-Tube (QFT-GIT) [34]. QFT-Plus has the potential to indicate recent infection and disease activity, by measuring the level of immune response. A recent meta-analysis found that QFT-Plus is a more sensitive test than QFT-GIT for detecting *M.tb* infection [35]. In 2019, Lionex GmbH introduced another new IGRA test (LIOFeron TB/TBI test), that may have higher sensitivity than the QFT-Plus assay [36,37]. The C-Tb assay is a novel skin test using the ESAT-6 and CFP-10 antigens, classically used in IGRA tests. This assay aims to combine the operational advantages of TST with the performance characteristics of IGRA. The C-Tb assay performed better than the TST in BCG-vaccinated individuals, had high concordance with the QFT-GIT test and was found to be safe in PLHIV and children under 5 years of age, with a similar positivity rate to the QFT test [38]. According to the WHO, this new type of *M.tb* antigen-based skin test is cost-effective or cost-saving as compared to TST and IGRA [39]. However, new IGRAs are more specific than new TST (C-Tb assay), as they detect only TB-specific responses and do not cross-react with other mycobacteria and BCG. Additionally, IGRAs seem to be less affected by immunosuppression because they are able to measure a broader range of immune responses [40]. Ultimately, the choice of test will depend on the clinical context and the availability of the test as well as the local epidemiology of TB (high- or low-TB incidence, proportion of BCG-vaccinated population) when selecting a diagnostic test for TBI [41].

### 3.2. Blood Biomarkers for Incipient TB 

The key purpose of diagnosing TBI is to identify persons who are most likely to progress to active TB [42]. Incipient TB is an attractive target for the development of new diagnostics. Whole blood biomarkers that can best predict the risk of progression to active TB are being tested using RNA sequencing in the blood of exposed persons, and studies identified genetic signatures for a risk of progression to TB disease within 6 to 12 months [43,44,45,46]. Recently, a three-gene signature set was shown to distinguish active TB from TBI and to correctly classify 91.5% of individuals (compared to 80–85% for previous tested signatures [47,48]). The diagnostic performance of this three-gene signature was tested in a clinical cohort of 147 subjects with suspected active TB, and the sensitivity and specificity for active TB were 82.4 and 92.4%, respectively [44] (compared to 71 and 89% for former signatures [47]).

Despite these advances, currently no diagnostic test can accurately detect TBI, or distinguish subclinical or early clinical disease from TBI, nor identify TBI due to drug-resistant strains of *M.tb* [49].

### 3.3. Tuberculosis Preventive Treatment

For more than 50 years, isoniazid has been the main drug used for TPT. However, the 6- to 9-month isoniazid regimen has shown low acceptance and completion rates, due to the long treatment duration, poor treatment adherence and poor tolerability [50,51,52].

Over the last decade, new treatments have been tested and proposed for TPT. Shorter regimens using rifampicin or rifapentine (a long-acting rifamycin), have been shown to be as effective as isoniazid-based regimens, with higher completion rates and better safety [53,54,55]. A daily dose of rifampicin for 4 months (4R) [53] or 12 weeks of weekly isoniazid and rifapentine (3HP) [54] were not inferior to 9 months of isoniazid for the prevention of active TB in adults and children. Both regimens showed better completion rates and fewer serious adverse events, particularly hepatitis. More recently, a regimen combining isoniazid and rifapentine daily for one month (1HP) [55] was found not inferior to 9 months of isoniazid with a higher completion rate in adults and adolescents [56]. Based on these results, the WHO issued in 2020 new recommendations for the provision of TPT in adults, adolescents and children at risk, either as living with HIV or as household contacts of people with bacteriologically confirmed pulmonary TB. These are outlined in Table 1 and the various regimens in Table 2 [23].

In selected high-risk household contacts of patients with MDR-TB, preventive treatment may be considered based on an individualized risk assessment and a sound clinical justification [23].

## 4. Cost-Effectiveness of TBI Prevention

A decision analysis model based on a cohort of 10,000 adults assessing all the WHO-recommended TPT regimens reported that, compared with no TPT, 3HP, 4R, 3RH, 9H and 6H reduced costs (costs of medications, medical supplies, medical personnel time and diagnostic procedures) and TB-related disabilities, and 3HP was the most cost-effective [57].

A systematic review and meta-analysis of costs, risks, benefits and impacts of TPT among PLHIV showed that TPT was overall cost-effective for preventing active TB as compared to no TPT at all [58]. Out of the sixty-one studies included, forty-five evaluated 6 to 12 months of daily isoniazid and nine considered rifamycin-based regimens. No TPT regimen was substantially more cost-effective at averting active TB than any other [58]. 

Considering the various regimens, 3HP appears as a cost-effective alternative to isoniazid preventive therapy in high-burden countries, but this is largely dependent on the price of rifapentine, local willingness to pay and the capacity to obtain a high completion rate (>85%) [59,60]. In high-income low-TB-burden countries, such as the United States, the 3HP regimen seems to be cost-effective when compared to 9H, especially if the cost of rifapentine decreases, adherence is maintained and the treatment is targeted towards individuals at high risk of progression to active TB [61].

Modelling studies also provided insights on potential cost-effectiveness of a new, shorter regimen. Considering the populations of PLHIV and HHCs in two countries representing two distinct epidemiological contexts, Brazil (low-TB incidence and low prevalence of HIV infection and rifampicin-resistant TB) and South Africa (high-TB incidence and high prevalence of HIV infection and rifampicin-resistant TB), a regimen meeting minimal requirements (3-month duration and 80% efficacy) was highly cost-effective, and even cost-saving, compared with expanding 6H [62]. In the same study, an optimal regimen (1-month duration and 100% efficacy) showed further cost-saving and health gains, primarily due to improved efficacy.

Lastly, a systematic review estimated that targeting contacts of MDR-TB cases with preventive therapy led to a 90% reduction in the incidence of MDR-TB. The cost-effectiveness was highest using a moxifloxacin/ethambutol combination regimen compared to other MDR-TB preventive regimens (pyrazinamide/ethambutol, moxifloxacin monotherapy, moxifloxacin/pyrazinamide, moxifloxacin/ethionamide and moxifloxacin/PA-824) [63].

## 5. Programmatic Challenges

Programmatic management of TBI is emerging as a critical component of TB control programs at the country level with the view to end TB globally [7]. It refers to the identification of TBI, early detection of active TB and prescription of TPT after ruling-out active TB in people at high risk of developing active TB such as recently exposed people, people with affected/suppressed immunity and migrants from high-TB incidence countries. It requires input from different components or technical units responsible for TB prevention and control, including the detection of individuals with TBI, treatment, provision of TPT, surveillance, monitoring and evaluation of the program performance [41].

### 5.1. TBI Screening

Optimizing screening approaches and making them accessible and available to people is one of the greatest challenges in enhancing TPT implementation [7]. The lack of a largely available and accessible test for TBI diagnosis with high sensitivity and specificity in resource-limited settings, and the absence of a highly reliable test to rule out TB are major challenges to decide in whom and when to start TPT. The WHO suggests a series of algorithms associating various combinations of symptoms, chest X-ray and rapid nucleic acid amplification tests to rule out TB [64]. However, low access to these tests, especially chest X-ray, and operational difficulties in the routine use of TST (storage of tuberculin, need of two visits, risk of false positive in BCG vaccinated people and negative results in immune-compromised people) and IGRA (costs and need of laboratory) [65] represent a challenge for programs, particularly in low-resource settings. Therefore, in high-TB incidence and low-resource settings, the WHO recommends initiating TPT in high-risk groups (PLHIV and young child contacts) without the need to confirm TBI and to rule out active TB based on a TB symptom screening when a chest X-ray is not available.

### 5.2. Initiating TB Preventive Therapy

The implementation of TPT, although improved in many countries and amongst certain risk groups, has still not reached the desired levels to achieve the End-TB strategy goals. Indeed, the reported proportion of PLHIV, HHCs < 5 years and HHCs ≥ 5 years placed on TPT between 2018 and 2021 were 10.3 million (>100% of target), 1.6 million (40% of target) and 0.6 million (3% of target), respectively. Various challenges have been identified for both the initiation and completion of TPT:-TPT priority, drug stock-outs or access problems: In TB-endemic areas where resources are limited, the focus is more on treating than preventing TB [66]. Fear of drug stock-outs, especially in peripheral centers, had been identified as the main cause of non-initiation of isoniazid preventive therapy [67]. In addition, operational and logistical issues may also lead to limited access to TBI screening and to TPT [68];-Lack of qualified human resources: The lack of qualified health care workers or the absence of training on how to prescribe TPT and convince asymptomatic individuals to undertake TBI screening or to take treatment may lead to distrust in health care workers [69,70];-Approach to screening and TPT: Routine screening for TBI may create anxiety and fear of stigma for individuals who do not feel sick. The WHO recommends decentralized, family-centered and integrated models of care to deliver services to children, adolescents and adults exposed to TB [71];-Diagnostic of TBI and active TB: The inherent limitations in the currently available tests for diagnosis of TBI as well as those for ruling out active TB and their limited availability in resource-poor countries are hampering wide programmatic management of TBI and may lead to the emergence of drug-resistance if TPT is not appropriately prescribed [72];-Availability of shortened regimens: The current high cost of rifapentine and the limited access to appropriate drug formulations, especially the dispersible rifapentine for infants, fixed-dose combinations for HP [41] and single-dose of rifampicin for 4R [23] are a further limit to the wide provision of TPT;-Monitoring and evaluation: Few low- and middle-income countries (LMICs) have an established system for monitoring and evaluating the implementation of TPT [73]. In addition, the reporting requirements by different donor agencies, which are not harmonized with international and national indicators, may contribute to unnecessary program overload [73].

## 6. Further Research and Future Direction

Improved point-of-care diagnostic tests that can reliably distinguish between TBI and active TB, combined with the capacity to assess the risk of active TB development would certainly be a major asset for TB prevention on a global scale. IGRA testing offers better potential to confirm and treat TBI than TST, but implementation remains limited (due to the lack of appropriate laboratory equipment and technical expertise) [74]. Further research is needed to better distinguish between TBI and active TB and to identify predictive biomarkers of disease progression. A recent pooled meta-analysis has reported that current candidate biomarkers have the ability to accurately reflect active TB risk and meet the WHO target product profile (including minimum sensitivity and specificity of 75%, and optimal sensitivity and specificity of 90%) mainly when considering a short-term risk of TB (period of 3 to 6 months after TBI) [46]. 

Along with improved diagnostics, there is a need to continue the search for a shorter or ultra-short (<1 month), safe and well-tolerated TPT that can be administered to all patients without concern for drug–drug interactions. While waiting for the results of the currently conducted trials, uncertainty persists regarding the optimal TPT strategies in communities with high rates of isoniazid- or fluoroquinolone-resistance [41]. Table 3 summarizes the current studies on TPT with new drugs such as bedaquiline, pretomanid or delamanid. Of note, we found studies in high-income settings as well as in low- and middle-income settings and all populations were covered (PLHIV, LGBT, MDR-TB). In addition, many ongoing studies are about shorter or ultra-short treatment with different administration strategies (daily 1HP and 3 times/week 1 HP, for example).

It would be helpful to consider the potential impact of different TBI management strategies on different population groups, such as those at high risk for progression to active TB in high-income countries or LMICs, those with limited access to health care and those in LMICs where the burden of TB is the highest.

In the meantime, using the currently available tools more efficiently and better targeting resources to those most at risk of active TB development or reactivation would go a long way toward reducing the global TB burden [75].

## 7. Conclusions

Tuberculosis infection is a complex and heterogeneous condition resulting from the interaction between the organism and the host immune response [8]. TBI affects one-quarter of the world’s population and constitutes the reservoir of active TB. Meeting the goals of the WHO End-TB strategy requires a multipronged approach that includes the systematic screening of high-risk populations, and targeted provision of preventive therapy with an effective monitoring and evaluation system. The global scale-up of TPT, especially in LMICs, requires wide access to diagnostic and prevention tools to permit the wide scale-up of this cornerstone strategy for ending TB globally, through adequate funding, and, more importantly, strong political commitment.

## Figures and Tables

**Figure 1 pathogens-12-00362-f001:**
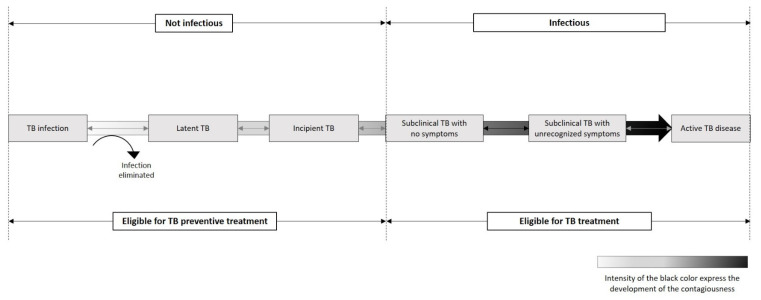
Spectrum of TB infection.

**Table 1 pathogens-12-00362-t001:** WHO guidelines on TB preventive therapy (source: WHO guidelines on Tuberculosis preventive treatment [23]).

People living with HIV
Adults and adolescents living with HIV who are unlikely to have active TB should receive TPT. Treatment should also be given to those on antiretroviral treatment, to pregnant women and to those who have previously been treated for active TB, irrespective of the degree of immunosuppression and even if TBI testing is unavailable.
Infants aged < 12 months living with HIV who are in contact with a person with TB and who are unlikely to have active TB should receive TPT.
Children aged ≥ 12 months living with HIV who are unlikely to have active TB should be offered TPT and care if they live in a setting with high TB transmission, regardless of contact with TB.
All children living with HIV who have successfully completed treatment for active TB may receive TPT.
**Household contacts (regardless of HIV status)**
Children aged < 5 years who are HHCs of people with bacteriologically confirmed pulmonary TB and who are found not to have active TB should be given TPT even if TBI testing is unavailable.
Children aged ≥ 5 years, adolescents and adults who are HHCs of people with bacteriologically confirmed pulmonary TB who are found not to have active TB may be given TPT.
In selected high-risk HHCs of patients with MDR-TB, TPT may be considered based on individualized risk assessment and a sound clinical justification.

**Table 2 pathogens-12-00362-t002:** TB preventive therapy recommended by the WHO (source: WHO guidelines on Tuberculosis preventive treatment [23]).

Medicines	Isoniazid 6H	Isoniazid + Rifapentine 3HP	Isoniazid + Rifampicin 3HR	Rifampicin 4R	Isoniazid + Rifapentine 1HP	Isoniazid 36H
**Duration (months)**	6	3	3	4	1	36
**Interval**	Daily	Weekly	Daily	Daily	Daily	Daily
**Indication**	All ages; child-friendly formulation available^£^; preferred in HIV+ children on LPV-RTV, NVP or DTG	≥2 years; no child-friendly formulation available	All ages; child-friendly formulation available and recommended up to 25 kg weight	All ages; no child- friendly formulation available; no formulation available for infants < 8 kg weight	>12 years; no rifapentine dosing available until 13 years of age	Adolescents and adults living with HIV
**Pregnant women**	Safe for use *	Not known	Safe for use *^$^	May be safe, although no safety or efficacy data available specifically in this population^$^	Not known	_

DTG = dolutegravir, H = isoniazid, LPV–RTV = lopinavir-ritonavir, NVP = nevirapine, P = rifapentine, R = rifampicin. ^£^ Available also in combination with pyridoxine and cotrimoxazole for people living with HIV. * One randomized trial has shown increased risk of poor birth outcomes for mothers taking isoniazid during pregnancy; however, several other studies have shown benefits of isoniazid preventive treatment; hence caution is required. ^$^ Bleeding attributed to hypoprothrombinaemia has been reported in infants and mothers following the use of rifampicin in late pregnancy. Vitamin K is recommended for both the mother and the infant postpartum if rifampicin is used in the last few weeks of pregnancy.

**Table 3 pathogens-12-00362-t003:** Ongoing clinical trials on tuberculosis preventive treatment.

Study (Reference)	Problem or Goal	Location	Design, Phase, Effective (n)	Treatment Strategy
Tuberculosis Preventive Therapy Among Latent Tuberculosis Infection in HIV-infected Individuals (NCT03785106) https://clinicaltrials.gov/ct2/show/NCT03785106 (accessed on 3 November 2022).	Treatment-shortening for TBI in HIV-infected patients	Thailand	Multicenter, open-label, randomized clinical trial Phase III n = 2500	4-week daily INH/RPT regimen (1HP) versus a 12-week INH/RPT regimen (3HP)
SCRIPT-TB (NCT03900858) https://clinicaltrials.gov/ct2/show/NCT03900858 (accessed on 3 November 2022).	Efficacy and safety of 1RPT/INH in preventing TBI	China	Open-label, randomized clinical trial Phase III n = 566	1-month (3 times/week = 12 doses) rifapentine and isoniazid (1RPT/INH) versus a 3-month weekly rifapentine and isoniazid regimen (3RPT/INH)
SCRIPT-LGTB (NCT04528277) https://clinicaltrials.gov/ct2/show/NCT04528277?term=rifapentine&draw=2 (accessed on 3 November 2022).	Efficacy and safety of 1RPT/INH in preventing latent genital TB preceding IVF among adult women with and without latent genital TB and experiencing recurrent implantation failure	China	Open-label, non-randomized clinical trial Phase III n = 1050	1-month (3 times/week = 12 doses) rifapentine and isoniazid (1RPT/INH) versus no treatment
PHOENIx MDR-TB (NCT03568383) https://clinicaltrials.gov/ct2/show/NCT03568383 (accessed on 3 November 2022).	Compare the efficacy and safety of 26 weeks of DLM for preventing confirmed or probable active TB among high-risk household contacts of adults with MDR-TB	Botswana, Brazil, Haiti, India, Kenya, Peru, Philippines, South Africa, Tanzania, Thailand, Uganda, Zimbabwe	Multicenter, open-label, randomized clinical trial Phase III n = 5610	26 weeks of DLM versus 26 weeks of INH
ASTERoiD/TBTC Study 37 (NCT03474029) https://clinicaltrials.gov/ct2/show/record/NCT03474029?term=Asteroid (accessed on 3 November 2022).	Compare the safety and effectiveness of a novel short 6-week regimen of daily rifapentine among people ≥12 years of age with positive TST or IGRA and at high risk of disease progression	Canada, United States	Multicenter, open-label, randomized clinical trial Phase III n = 3400	6-week P vs. rifamycin-based standard-of-care regimens (3HP, 4R or 3HR)
SDR Risk Study (NCT04094012) https://clinicaltrials.gov/ct2/show/record/NCT04094012?term=rifapentine&draw=6 (accessed on 3 November 2022).	Compare incidence rate of systemic drug reactions under 3HP and 1HP regimen for latent tuberculosis infection treatment	Taiwan	Pragmatic open-label, multicenter randomized control trial Phase III n = 490	3-month HP versus 1-month HP
2R2 (NCT03988933) https://clinicaltrials.gov/ct2/show/record/NCT03988933?recrs=ab&rslt=Without&type=Intr&cond=TB&phase=123&sort=nwst&draw=2 (accessed on 3 November 2022).	Determine if rifampin at double or triple the standard dose for 2 months is as safe and effective as the standard dose	Canada, Vietnam	Multicenter, randomized, partially blind, controlled trial Phase IIb n = 1359	High-dose R (20 or 30 mg/kg) taken daily for 2 months versus 4-month R
TPT and Rheumatic Disease (ChiCTR1800018242) https://www.chictr.org.cn/showprojen.aspx?proj=30532 (accessed on 3 November 2022).	Evaluate the effectiveness, safety and compliance of different preventive anti-tuberculosis treatments in rheumatic patients at high risk of active tuberculosis	China	Multicenter, open-label, randomized clinical trial Phase IV n = 500	3 months of RPT/INH versus 9 months of INH
PROTID (NCT04600167) https://clinicaltrials.gov/ct2/show/record/NCT04600167?term=rifapentine&draw=5 (accessed on 3 November 2022).	Safety and efficacy of 3HP vs. placebo to prevent TB in people with diabetes	Uganda, Tanzania	Multicenter, randomized, double blind, placebo-controlled trial Phase III n = 3000	One weekly 3 months of RPT/INH versus placebo
Ultra Curto (NCT04703075) https://clinicaltrials.gov/ct2/show/record/NCT04703075?term=rifapentine&draw=2 (accessed on 3 November 2022).	Treatment success and safety of 1HP vs. 3HP among HIV-negative adult and adolescent HHCs and documented conversion within 2 years	Brazil	Multicenter, open-label, randomized clinical trial Phase IV n = 500	

MDR-TB: multidrug-resistant tuberculosis; DLM: delamanid; BCG: Bacille Calmette Guerin; RPT or P: Rifapentine; INH or H: Isoniazide; TBI: tuberculosis infection; LTBI: latent tuberculosis infection; R: Rifampicin; IVF: in vitro fertilization; HHC: household contacts.

## Data Availability

Data is contained within the article. The data presented in this study are available in insert articles.

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
