# Peer review of "Management of Tuberculosis Infection: Current Situation, Recent Developments and Operational Challenges"

_pathogens, 2023, doi:10.3390/pathogens12030362_

Round 1

Reviewer 1 Report

This is a very nice piece discussing TB infection and prevention. The strengths of this paper were the sections outlining new diagnostic tools and novel shortened TPT strategies under study.  

Suggest the abstract, and title are restructured to reflect the aforementioned focus and strength of the manuscript.

As mentioned above, the title does not really accurately reflect the manuscript content, the authors do not make recommendations and very little is discussed on cost-effectiveness, or differences in high and low burden countries. Would suggest considering a title such as the “current and future landscape of TB infection prevention:”.

The discussion outlining newer diagnostic tests for TB infection was very nice, and could be expanded even further for readers to understand better where different testing options may be employed.

Figure 1 is very helpful for readers and represents the TB infection and disease spectrum. I suggest that since it is proposed as a spectrum, the figure would be better served if the blocks were connected with shading or internal arrows. Likewise, it would be helpful if the infectious period was not denoted by a single line but a gradient that shows the development of infectiousness is likely slightly different between individuals.

Minor comments

All sections seem to be labeled 1.1 making it difficult to follow.

1.1 Global Burden section lines 76-80:  The first paragraph appears to be one long sentence. Authors refer to “that model” but reference a new citation, therefore it is unclear whether the referenced model has been mentioned or introduced previously.

Line 94 refers to risk in terms of percentages, clarify whether these are annual risk or cumulative lifetime risk.

Line 104-109 the terms risk and incidence are used interchangeably here, but the critical issue is presenting incidence rates as percentages, this is not correct.

TBI and Covid pandemic: While it is well understood by the TB community, that TB incidence did not actually reduce during the pandemic, this section would be strengthened by a sentence citing the recent estimates of TB cases highlighting the growing gap between new cases and diagnosed cases, particularly if targeting this paper to a non TB expert audience.

There is a disconnect with earlier sections when authors indicate TPT can be initiated without confirming TBI and section TBI diagnosis where authors say:” tools are needed to identify patient with TBI reliably for rational provision of TPT”.

Lime 150 should be “theoretically”

Almost all of cost-effectiveness section is dedicated to studies assessing cost-effectiveness among PLHIV but this isn’t explicitly discussed and may not be clear to the casual reader.  Suggest a couple sentences are added to clearly discus CE among non PLHIV populations, even if to say evidence is limited. There is a substantial amount of literature around CE of shortened regimens in low TB incidence high income settings and this is not mentioned.

Would be helpful to define what authors mean by programmatic management of TB (line 239 and 240) before delving into smaller-components.

Line 165: Something cannot be as cost-saving as its comparator I am not sure the authors are interpreting the cost-effectiveness discussion from the cited report correctly here.

The section on programmatic challenges around TBI screening seems incomplete, would be helpful to at least outline or mention challenges beyond availability of diagnostic tools including challenges with hard to reach areas, distrust of health care workers, apathy or stigma towards screening, lack of expertise and or infrastructure required beyond diagnostic tools, sustained funding, etc.

Line 262 IPT introduced without defining it first.

Lines 264 and on feel a bit like a list, suggest rephrasing this section and grouping similar challenges together such as ‘availability of shortened regimens’ and ‘infrastructure challenges’ so it all ties together.

Table 2 seems to be a major element of this paper, a more in depth discussion here would be helpful.

Author Response

Answers to reviewers' comments (in red)

First of all, we would like to thank all the reviewers for taking the time to review our manuscript and for their valuable comments on our paper.

We answered their questions and comments as best we could. The required clarifications have been made in the revised manuscript when necessary.

Reviewer #1

This is a very nice piece discussing TB infection and prevention. The strengths of this paper were the sections outlining new diagnostic tools and novel shortened TPT strategies under study.  

Suggest the abstract, and title are restructured to reflect the aforementioned focus and strength of the manuscript.

Response: The abstract have been modified to take into account the comment of the reviewer. See lines 10-21

As mentioned above, the title does not really accurately reflect the manuscript content, the authors do not make recommendations and very little is discussed on cost-effectiveness, or differences in high and low burden countries. Would suggest considering a title such as the “current and future landscape of TB infection prevention:”.

Response: We agree with the reviewer. The title has been modified. The new title is: “Management of tuberculosis infection: current situation, recent developments and operational challenges”

The discussion outlining newer diagnostic tests for TB infection was very nice, and could be expanded even further for readers to understand better where different testing options may be employed.

Response: A paragraph has been added to the text to take account of this comment. See lines 171-177.

Figure 1 is very helpful for readers and represents the TB infection and disease spectrum. I suggest that since it is proposed as a spectrum, the figure would be better served if the blocks were connected with shading or internal arrows. Likewise, it would be helpful if the infectious period was not denoted by a single line but a gradient that shows the development of infectiousness is likely slightly different between individuals.

Response: Figure 1 has been modified accordingly.

Minor comments

All sections seem to be labeled 1.1 making it difficult to follow.

Response: Modifications of labeling have been made

  • Global Burden section lines 76-80:  The first paragraph appears to be one long sentence. Authors refer to “that model” but reference a new citation, therefore it is unclear whether the referenced model has been mentioned or introduced previously.

Response: The clarification has been made and the first paragraph of Global burden has been substantially modified. See lines 76-82

Line 94 refers to risk in terms of percentages, clarify whether these are annual risk or cumulative lifetime risk.

Response: It is a proportion of individual at risk of progression to TB disease. We added it in the text (lines 97-99)

Line 104-109 the terms risk and incidence are used interchangeably here, but the critical issue is presenting incidence rates as percentages, this is not correct.

Response: Here it is a 2-year cumulative incidence that include prevalent and incident TB cases in the first 2-years of follow-up from prospective cohort studies. Clarifications have been made in the text. See lines 111-116.

TBI and Covid pandemic: While it is well understood by the TB community, that TB incidence did not actually reduce during the pandemic, this section would be strengthened by a sentence citing the recent estimates of TB cases highlighting the growing gap between new cases and diagnosed cases, particularly if targeting this paper to a non TB expert audience.

Response: The comment is correct, during the COVID-19 pandemic, it has been observed that TB incidence didn't reduce. We added information about importance of this growing gap and some figures from the WHO Global TB Report 2022. See lines 141-143.

There is a disconnect with earlier sections when authors indicate TPT can be initiated without confirming TBI and section TBI diagnosis where authors say:” tools are needed to identify patient with TBI reliably for rational provision of TPT”.

Response: Due to operational constraints with poor access to TBI test and chest X-ray to rule out TB disease, in high TB incidence and resource limited countries, for groups at high risk of developing the disease after exposure such as PLHIV and young child contacts, WHO recommends that TPT can be initiated without confirming TBI and that TB disease can be ruled out using symptom screening with chest X-ray. The sentence that caused the confusion has been deleted and modifications have been made in the text (lines 148-157).

Line 150 should be “theoretically”

Response: Correction made (line 155)

Almost all of cost-effectiveness section is dedicated to studies assessing cost-effectiveness among PLHIV but this isn’t explicitly discussed and may not be clear to the casual reader.  Suggest a couple sentences are added to clearly discus CE among non PLHIV populations, even if to say evidence is limited. There is a substantial amount of literature around CE of shortened regimens in low TB incidence high income settings and this is not mentioned.

Response: Information have been added in the text. See lines 227-230 and 239-242.

Would be helpful to define what authors mean by programmatic management of TB (line 239 and 240) before delving into smaller-components.

Response: These information have been added in the text. See lines 259-266

Line 165: Something cannot be as cost-saving as its comparator I am not sure the authors are interpreting the cost-effectiveness discussion from the cited report correctly here.

Response: Modifications have been made in the text. See lines 170-171

The section on programmatic challenges around TBI screening seems incomplete, would be helpful to at least outline or mention challenges beyond availability of diagnostic tools including challenges with hard to reach areas, distrust of health care workers, apathy or stigma towards screening, lack of expertise and or infrastructure required beyond diagnostic tools, sustained funding, etc.

Response: Clarifications and additions have been made. See lines 283-316.

Line 262 IPT introduced without defining it first.

Response: Definition has been added (line 292)

Lines 264 and on feel a bit like a list, suggest rephrasing this section and grouping similar challenges together such as ‘availability of shortened regimens’ and ‘infrastructure challenges’ so it all ties together.

Response: Modifications have been made in the text. See lines 283-316

Table 2 seems to be a major element of this paper, a more in depth discussion here would be helpful.

Response: Discussion points have been added. See lines 335-338.

Reviewer 2 Report

The manuscript is overall well-written and systematically presented. The review article gives a sound idea of scientific facts related to TB infection prevention. Figure and data table contribute fair amount of suppportive data to the idea presented here. However, the authors need to look more carefully for minor English mistakes, like spelling errors, grammatical errors, and poor sentence framing. For e.g.:

1. line no. 33-34 is grammatically incorrect.

2. line no. 40 has a spelling error, 'miner' should be corrected to minor.

3. line no. 42: poorly framed sentence.

4. line no. 77: punctuation is missing after '31%'

Similarly, line no. 118, 131, 150 also need to be corrected. There could be more errors in the manuscript. Please go through the entire manuscript carefully and rectify wherever required. Also, make sure to avoid repetitive use of same phrases.

5. Abbreviation LMICs has not been mentioned in full anywhere in manuscript. Please de-abbreviate at first instance of appearance.

6. Figure 1: Authors should represent latent TB as another arm post infection that may/ may not lead to incipient infection, instead of showing it as a transient phase before incipient infection. The reason is that Latent TB may not ever develop into incipient TB and could remain dormant life-long.

7. Why do authors use double headed arrows everywhere in Fig 1? Does it indicate bidirectionality at all phases of Tuberculosis manifestation or spontaneous reversibility? If not, then it is counterintuitive. Authors might want to replace it with unidirectional arrows, wherever applicable.  

8. It would be helpful for the general audience if the authors can include some data on which countries are high burden and low burden in context of TB infection.

Author Response

Answers to reviewers' comments (in red)

First of all, we would like to thank all the reviewers for taking the time to review our manuscript and for their valuable comments on our paper.

We answered their questions and comments as best we could. The required clarifications have been made in the revised manuscript when necessary.

Reviewer #2

The manuscript is overall well-written and systematically presented. The review article gives a sound idea of scientific facts related to TB infection prevention. Figure and data table contribute fair amount of supportive data to the idea presented here. However, the authors need to look more carefully for minor English mistakes, like spelling errors, grammatical errors, and poor sentence framing. For e.g.:

  1. Line no. 33-34 is grammatically incorrect.

Response: The sentence has been modified. See lines 32-33.

  1. Line no. 40 has a spelling error, 'miner' should be corrected to minor.

Response: Modifications have been made. See lines 39-42

  1. Line no. 42: poorly framed sentence.

Response: The sentence has been reformulated. See lines 39-42.

  1. Line no. 77: punctuation is missing after '31%'

Response: Indeed, there is a missing full stop after "31%". Correction made. See line79

Similarly, line no. 118, 131, 150 also need to be corrected. There could be more errors in the manuscript. Please go through the entire manuscript carefully and rectify wherever required. Also, make sure to avoid repetitive use of same phrases.

Response: Modifications have been made and the manuscript has been revised substantially.

  1. Abbreviation LMICs has not been mentioned in full anywhere in manuscript. Please de-abbreviate at first instance of appearance.

Response: Modifications have been made.

  1. Figure 1: Authors should represent latent TB as another arm post infection that may/ may not lead to incipient infection, instead of showing it as a transient phase before incipient infection. The reason is that Latent TB may not ever develop into incipient TB and could remain dormant life-long.

Response: Figure 1 has been modified.

  1. Why do authors use double headed arrows everywhere in Fig 1? Does it indicate bidirectionality at all phases of Tuberculosis manifestation or spontaneous reversibility? If not, then it is counterintuitive. Authors might want to replace it with unidirectional arrows, wherever applicable. 

Response: Indeed, double headed is to indicate bidirectionality at all phases of Tuberculosis manifestation or spontaneous reversibility.

  1. It would be helpful for the general audience if the authors can include some data on which countries are high burden and low burden in context of TB infection.

Response: This information has been added in the text. See lines 76-82.

Reviewer 3 Report

The authors have investigated an important theme regarding infection prevention. Tuberculosis is a world wide disease still of difficult management and that has been set to be dealt with by the WHO by the End-TB program. Therapy is now accessible and feasible thanks to the easy to acquire necessary antibiotics. Limitation to disease management is mainly due to screening in the industrialized countries and due to treatment in the developing ones. This review highlights the state of the art of TB management in regards to recommendations and cost-effectiveness in high and low burden countries. 

The study is on point, well thought of and overall decently written. Introduction is sufficient for the purpose. In chapter 1.1 efforts have been made for management in TB in desperate times of COVID in an difficult to detect and treat population (35206999). This should be addressed as well as special population of critical importance in industrialized countries (28482956). I also suggest the need for an immagine regarding TB management.

The scientific community due to the great burden of COVID 19 has set aside other calamities and this review sets to hopefully restart the management of some of these.

Author Response

Answers to reviewers' comments (in red)

First of all, we would like to thank all the reviewers for taking the time to review our manuscript and for their valuable comments on our paper.

We answered their questions and comments as best we could. The required clarifications have been made in the revised manuscript when necessary.

Reviewer #3

The authors have investigated an important theme regarding infection prevention. Tuberculosis is a world wide disease still of difficult management and that has been set to be dealt with by the WHO by the End-TB program. Therapy is now accessible and feasible thanks to the easy to acquire necessary antibiotics. Limitation to disease management is mainly due to screening in the industrialized countries and due to treatment in the developing ones. This review highlights the state of the art of TB management in regards to recommendations and cost-effectiveness in high and low burden countries. 

The study is on point, well thought of and overall decently written. Introduction is sufficient for the purpose. In chapter 1.1 efforts have been made for management in TB in desperate times of COVID in an difficult to detect and treat population (35206999). This should be addressed as well as special population of critical importance in industrialized countries (28482956). I also suggest the need for an immagine regarding TB management.

Response: We did not add this information as the article is on TB infection management and not TB disease. Although there are data on the impact of covid-19 on TB disease and some success story, there are no data about impact of COVID-19 on provision of TPT – only hypotheses that it has followed the decline in TB case management.

The scientific community due to the great burden of COVID 19 has set aside other calamities and this review sets to hopefully restart the management of some of these.

Round 2

Reviewer 3 Report

The authors have not made the required adjustments. All the more since the title has been adjusted ad the highlighted points must be addressed. 

"In chapter 1.1 efforts have been made for management in TB in desperate times of COVID in a difficult to detect and treat population (35206999). This should be addressed as well as special population of critical importance in industrialized countries (28482956)."

Citing the WHO reference is correct but if there is no consideration of new letterature, of the efforts made by other researchers, the review losses meaning. 

Author Response

Answers to reviewers' comments (in red)

First of all, we would like to thank all the reviewers for taking the time to review our manuscript and for their valuable comments on our paper.

We answered their questions and comments as best we could. The required clarifications have been made in the revised manuscript when necessary.

Reviewer #3

The authors have not made the required adjustments. All the more since the title has been adjusted ad the highlighted points must be addressed.

"In chapter 1.1 efforts have been made for management in TB in desperate times of COVID in a difficult to detect and treat population (35206999). This should be addressed as well as special population of critical importance in industrialized countries (28482956)."

Citing the WHO reference is correct but if there is no consideration of new literature, of the efforts made by other researchers, the review losses meaning.

Response: Thank for your comment. We have added this information in the text, as well as relevant literature. See lines 146-158.
